# In Vitro Evaluation of Wood Vinegar (Pyroligneous Acid) VOCs Inhibitory Effect against a Fungus-like Microorganism *Ovatisporangium* (*Phytopythium*) Isolate Recovered from Tomato Fields in Iran

Ali Chenari Bouket [1,*], Abolfazl Narmani [2], Alireza Tavasolee [3,†], Ghorban Elyasi [4,†], Akbar Abdi [5,†], Shahram Naeimi [6], Kasra Sharifi [7], Tomasz Oszako [8], Faizah N. Alenezi [9] and Lassaad Belbahri [10]

1 East Azarbaijan Agricultural and Natural Resources Research and Education Centre, Plant Protection Research Department, Agricultural Research, Education and Extension Organization (AREEO), Tabriz 5355179854, Iran

2 Department of Plant Protection, Faculty of Agriculture, University of Tabriz, Tabriz 5166614766, Iran; abolfazl.narmani2@gmail.com

3 East Azarbaijan Agricultural and Natural Resources Research and Education Center, Soil and Water Research Department, Agricultural Research, Education and Extension Organization (AREEO), Tabriz 5355179854, Iran; ar.tavasolee@areeo.ac.ir

4 East Azarbaijan Agricultural and Natural Resources Research and Education Center, Animal Sciences Research Department, Agricultural Research, Education and Extension Organization (AREEO), Tabriz 5355179854, Iran; gh.elyasi@areeo.ac.ir

5 East Azarbaijan Agricultural and Natural Resources Research and Education Center, Natural Resources Research Department, Agricultural Research, Education and Extension Organization (AREEO), Tabriz 5355179854, Iran; a.abdi@areeo.ac.ir

6 Biological Control Research Department, Iranian Research Institute of Plant Protection, Agricultural Research, Education and Extension Organization (AREEO), Tehran 1985813111, Iran; sh.naeimi@areeo.ac.ir

7 Plant Disease Research Department, Iranian Research Institute of Plant Protection, Agricultural Research, Education and Extension Organization (AREEO), Tehran 1985813111, Iran; ka.sharifi@areeo.ac.ir

8 Department of Forest Protection, Forest Research Institute in Sekocin Stary, 05-090 Raszyn, Poland; t.oszako@ibles.waw.pl

9 Marine Biodiscovery Centre, Department of Chemistry, University of Aberdeen, Old Aberdeen, Scotland AB24 3UE, UK; faizah.alenezi@abdn.ac.uk

10 Laboratory of Soil Biology, University of Neuchatel, 2000 Neuchatel, Switzerland; lassaad.belbahri@unige.ch

* Correspondence: a.chenari@areeo.ac.ir; Tel.: +98-413-244-3093; Fax: +98-413-244-2866

† These authors contributed equally to this work.

**Abstract:** Crop diseases and agricultural pests and pathogens are causing huge economic losses. The actual means for dealing with them involve the use of damaging chemical pesticides that harm the environment, threaten biodiversity, and undermine human health. This research was aimed at developing an environmentally friendly means to cope with emerging oomycete disease from tomato fields in the province of East-Azerbaijan. The oomycete disease causal agent was isolated and identified as *Ovatisporangium* sp. using a combination of morphological features and molecular methods. Six wood vinegars (pyroligneous acid) belonging to pine, pomegranate, pistachio, almond, walnut, and cypress were produced during this study and examined against *Ovatisporangium* sp. Their inhibition of volatile metabolites (VOCs) using different dilutions (1, 1/2, 1/4, 1/8, and 1/10) was assessed against the mycelial growth of *Ovatisporangium* sp. In vitro analysis demonstrated that pistachio, cypress, and almond dilution 1 (D 1) wood vinegar VOCs had the ability to stop the mycelial growth of *Ovatisporangium* sp. All other treatments including pine, walnut, and pomegranate with relevant dilutions significantly reduced the mycelial growth of *Ovatisporangium* sp. compared with the control ($p \leq 0.05$). Wood vinegar is therefore a potent means to cope with pathogenic infections and allows plant protection against oomycete diseases.

**Keywords:** *Ovatisporangium* (*Phytopythium*); in vitro; pyroligneous acid; VOCs; wood vinegar

## 1. Introduction

The application of synthetic pesticides including insecticides [1], fungicides [2], herbicides [3], nematicides [4], and acaricides [5] is a common strategy to manage weeds, pests, and the causal agents of plant disease. The side effects of these chemicals have negatively impacted the environment and ecological niches [6–10]. Synthetic pesticides have impacted human health and the environment [11]. They are among the top list of environmental toxicants having impacts on the soil [12], water [13], crops [14,15], and animals, including humans [16]. Nowadays, sustainable agriculture guidelines prohibit the use of numerous synthetic agrochemicals in crops and fruits production based on their damaging effects on the environment [17–20].

Thermo-pyrolysis is an environmentally friendly process used to produce organic materials such as tar, wood vinegar, and biochar (charcoal added to compost) [21]. Plant-derived bioproduct use in agriculture was historically documented in some regions, e.g., Greece and China [22,23]. Wood vinegar (pyroligneous acid (PA)) is an unpurified concentrated, extraordinary, oxygenated biomaterial extracted by the thermo-pyrolysis processes [24,25]. Wood vinegar is composed of more than 200 compounds, 20% of which is organic materials (e.g., phenol, alcohol, acids, and esters) and the remaining 80% is water [26–28]. Wood vinegar has two major benefits: (i) management of plant diseases caused by phytopathogens including fungi and bacteria and (ii) promotion of plant growth [29–31]. PA may significantly enhance the biodiversity of microbiota in soil, and improve soil biological, physical, and chemical conditions (improving seed germination, plant growth, fruit size, and vegetable quality). Therefore, it can be efficiently used as an organic fertilizer with pesticide properties [7,26,32]. Several woody plant biomasses, including Japanese cedar 'Sugi' [33], walnut [34], cherry [35], halophyte tree mangrove [36], small-flower chaste tree [37], beech [38], oak [39], birch [40], rosemary [41], eucalyptus [42], and bamboo, ref. [6] have been used for production of wood vinegar, with the final aim of increasing seedling germination and the development of crops such as lettuce, cucumber, chrysanthemum, and watercress [43,44].

The pesticidal effects of wood vinegar produced with mixed material was confirmed to have a mortality rate of more than 90% for PA-treated aphid (*Myzus persicae*) and mite (*Tetranychus urticae*) [44]. In Thailand, PA was used as a repellent against two important pests, snails and slugs [45], and caused approximately 95% mortality rate in aphid populations on eggplant [46]. Wood vinegar termicidal effectiveness was reported on Japanese termite *Reticulitermes speratus* [47]. Based on the presence of biomaterials, mainly phenols and organic acids, among other compounds, wood vinegar has a herbicidal effect on weeds propagules, especially in freshwater plants, e.g., *Potamogeton*, *Hydrilla*, and *Spartina* [48].

PA antimicrobial activity has not yet been extensively explored. The antibacterial effects of PA were assayed on some phytopathogenic bacteria, e.g., *Pectobacterium carotovorum* and *Xanthomonas campestris* pv. *citri* [49], *Ralstonia solanacearum* [33], *Agrobacterium tumefaciens* [45], and *Corynebacterium agropyri* [50]; and plant pathogenic fungi and fungal-like organisms, e.g., *Colletotrichum orbiculare*, *Valsa mali*, *Helminthosporium sativum*, *Cochliobolus sativus* [27] *Alternaria mali* [51], *Phytophthora infestans*, and *Phytophthora capsici* [52]. Chen et al. [53] reported a strong effect of wood vinegar against white rot (*Coriolus versicolor*) and brown rot fungi (*Gleophylum trabeum*). Saberi et al. [54] demonstrated that nonvolatile and volatile PA compounds had inhibitory effects on *Sclerotinia sclerotiorum* and *Rhizoctonia solani* in greenhouse-cultivated cucumber. Saberi et al. [55] also evaluated wood vinegar's impact on damping-off of cucumber, and showed that significant decreases in *Pythium aphanidermatum* and *Phytophthora drechsleri* mycelial growth occurred. Chuaboon et al. [56] explored wood vinegar's effect on rice diseases, and found that fungal causal agents such as *Curvularia lunata*, *Fusarium semitecum*, *Cercospora oryzae*, *Bipolaris oryzae*, and *Alternaria padwickii* were negatively affected. Additionally, Xu et al. [57] found that *Malus* sp. and *Pyrus* sp. wood vinegar components inhibited the mycelial growth of hiratake mushroom *Pleurotus ostreatus*. Given the lack of data concerning the effect of wood vinegar on oomycetes and its mechanism of action, we aimed in this study to explore the

in vitro inhibitory effect of volatile components of six trees' wood vinegar extracts against a *Phytopythium* isolate recovered from tomato fields.

## 2. Materials and Methods

### 2.1. Isolation

*Phytopythium* sp. isolates were recovered from tomato fields in the province of East-Azerbaijan, Iran (Figure 1). Diseased crown and root pieces of *Solanum lycopersicum* were surface-sterilized in $H_2O_2$ for 1 min, followed by incubation in hypochlorite sodium 4% for 5 min and 70% alcohol for 30 s [58]. Samples were then washed in sterile distilled water and dried on a Whatman towel for 5 min. Samples were placed on a semiselective oomycete medium (corn meal agar medium (CMA) amended with fluazinam + nystatin + ampicillin + rifampicin antibiotics) [59] and incubated at room temperature for 2–3 days. Cultures were purified using a hyphal tip technique in water agar (WA) media, and then preserved on CMA slant vials at 10 °C in the dark until use.

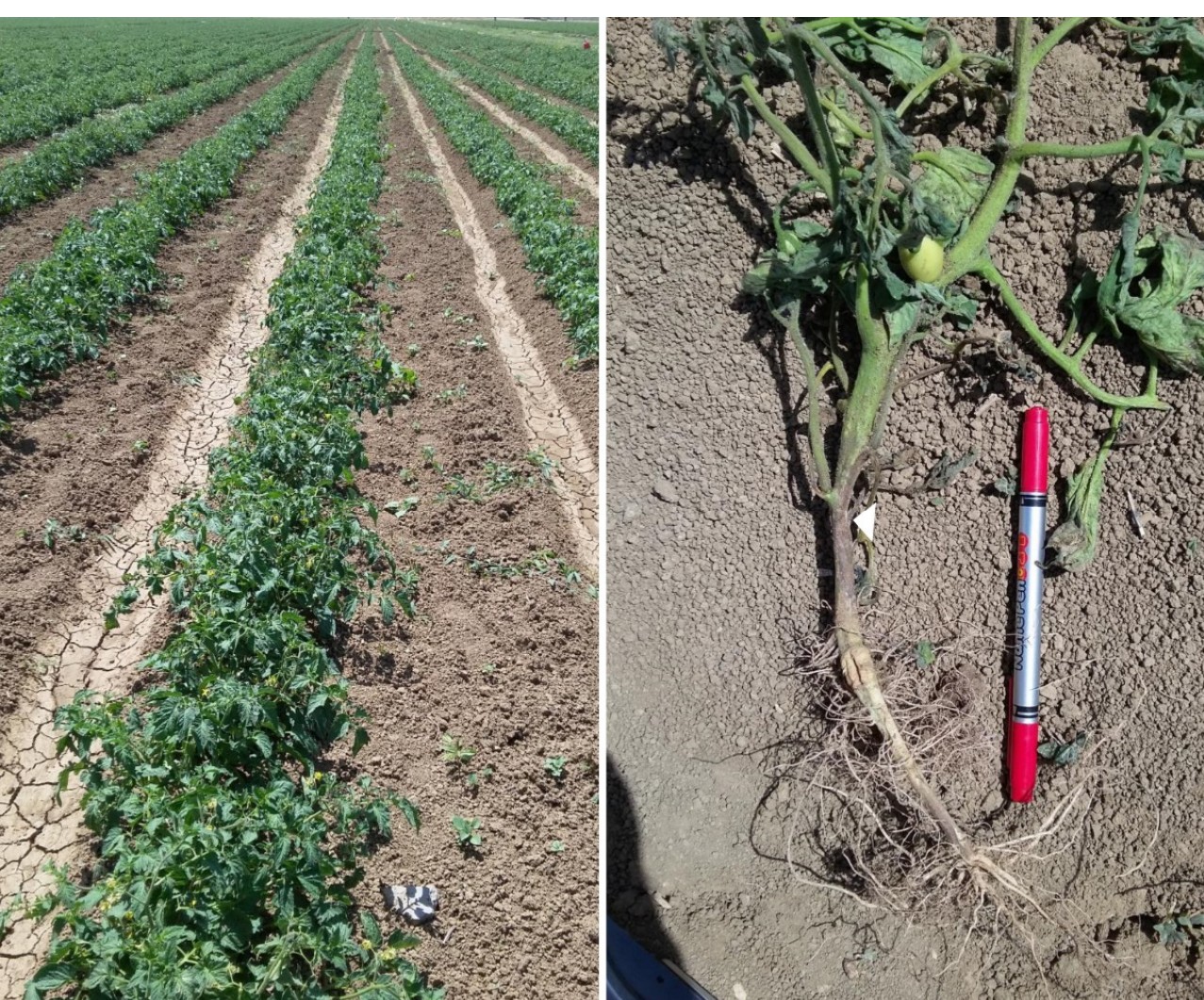

**Figure 1.** A tomato crop field (**left**) and oomycete-diseased plant seedling in early growing season with collar rot (white arrow) symptoms (**right**).

### 2.2. DNA Extraction and Amplification

DNA was extracted from oomycete mycelia cultivated on corn meal agar with the manual protocol described by Möller et al. [60]. The ITS-rDNA region was amplified with thermal cycling processes using universal ITS4 (TCCTCCGCTTATTGATATGC) and ITS5

(GGAAGTAAAGTCGTAACAAGG) primers [61]. All reactions were adjusted to 50 μL total volume, including 5 μL 10× Ex Taq buffer (20 mM Tris/HCl, pH 8.0, 100 mM KCl), 4 μL 2.5 mM dNTP, 0.5 μM of forward and reverse primers, 1.25 U Taq DNA polymerase (Takara Bio®), and 10 ng of DNA. Amplifications were carried out using a PerkinElmer 9700 thermal cycler (PerkinElmer®) machine using the following cycling profile: 95 °C for; 5 min followed by 30 cycles; denaturation step: 95 °C for 30 s; annealing step: 55 °C for 30 s; extension step: 72 °C for 1 min; and a final extension step: 72 °C for 7 min.

Purified amplicons were used for sequence analysis, and recovered sequences were deposited in GenBank [59].

### 2.3. Sequencing and Phylogeny

Amplicons were sequenced in both the 3′ and 5′ sides using amplification PCR primers and a BigDye Terminator v. 3.1 cycle sequencing kit (Applied Biosystems®, Waltham, MA, USA) following the manufacturer recommendations. They were then analyzed on a 3130×l Genetic Analyzer (Applied Biosystems®). Raw sequence files were edited using the SeqManII program (DNAStar®, Madison, WI, USA), and a consensus sequence was generated [62]. The consensus sequence for each genomic region was blasted against the NCBI's GenBank sequence database for detection of closest neighbor taxa. The sequences retrieved from GenBank together with the sequence provided in this paper were aligned using the multiple-sequence alignment Mega 6 program [63]. Trees were generated using the maximum likelihood (ML) method [64], with evolutionary distances computed using the Kimura 2-parameter model [65]. Bootstrap resampling of the data sets with 1000 replications was applied for branches support evaluation in the resulting trees [66].

### 2.4. Preparation of PA

Six trees (pine, pomegranate, pistachio, almond, walnut, and cypress) were selected in this study. Pyroligneous acids were extracted following the procedures described by Bridgwater et al. [67,68] and Mohan et al. [26].

### 2.5. In Vitro Evaluation of Wood Vinegar VOCs for Inhibition of Phytopythium

We poured 15 mL of potato dextrose agar (PDA) medium in one side of two-compartment petri dish plates and inoculated with 3 mm *Phytopythium* agar discs. On the other side of the plate, sterile distilled water was used as a control, and different PA dilutions (1, 1/2, 1/4, 1/8, 1/10) were poured. The plates were then sealed with Parafilm M® and incubated at room temperature [49]. After 2–3 days, oomycete growth (mm) was recorded, and the percentage of inhibition was estimated with the following formula [51]:

$$X = (A - B)/A \times 100 \tag{1}$$

where X is percentage of inhibition, A is the growth of fungi in the control petri plate, and B is the growth of fungi in each treated petri plate.

### 2.6. Statistical Analysis

The data statistical analysis was carried out using variance analysis (ANOVA) and, when significant effects were detected, the groups were compared with a post hoc Tukey's HSD test. The level of significance used for all statistical tests was 5% ($p < 0.05$). IBM SPSS Statistics v. 24 was used for the analysis.

### 3. Results

#### 3.1. Molecular Identification of Phytopythium Isolate

Different samples of collected pathogenic isolates were analyzed using the growth patterns on petri plates and morphological characteristics and subjected to DNA identification. The ITS-rDNA sequences of the specimens recovered from tomato (ON409943) were

identical and formed a clade with 82% bootstrap support with related *Phytopythium* species in the tree, including *Pp. delawarense*, *Pp. citrinum*, and *Pp. litorale* (Figure 2).

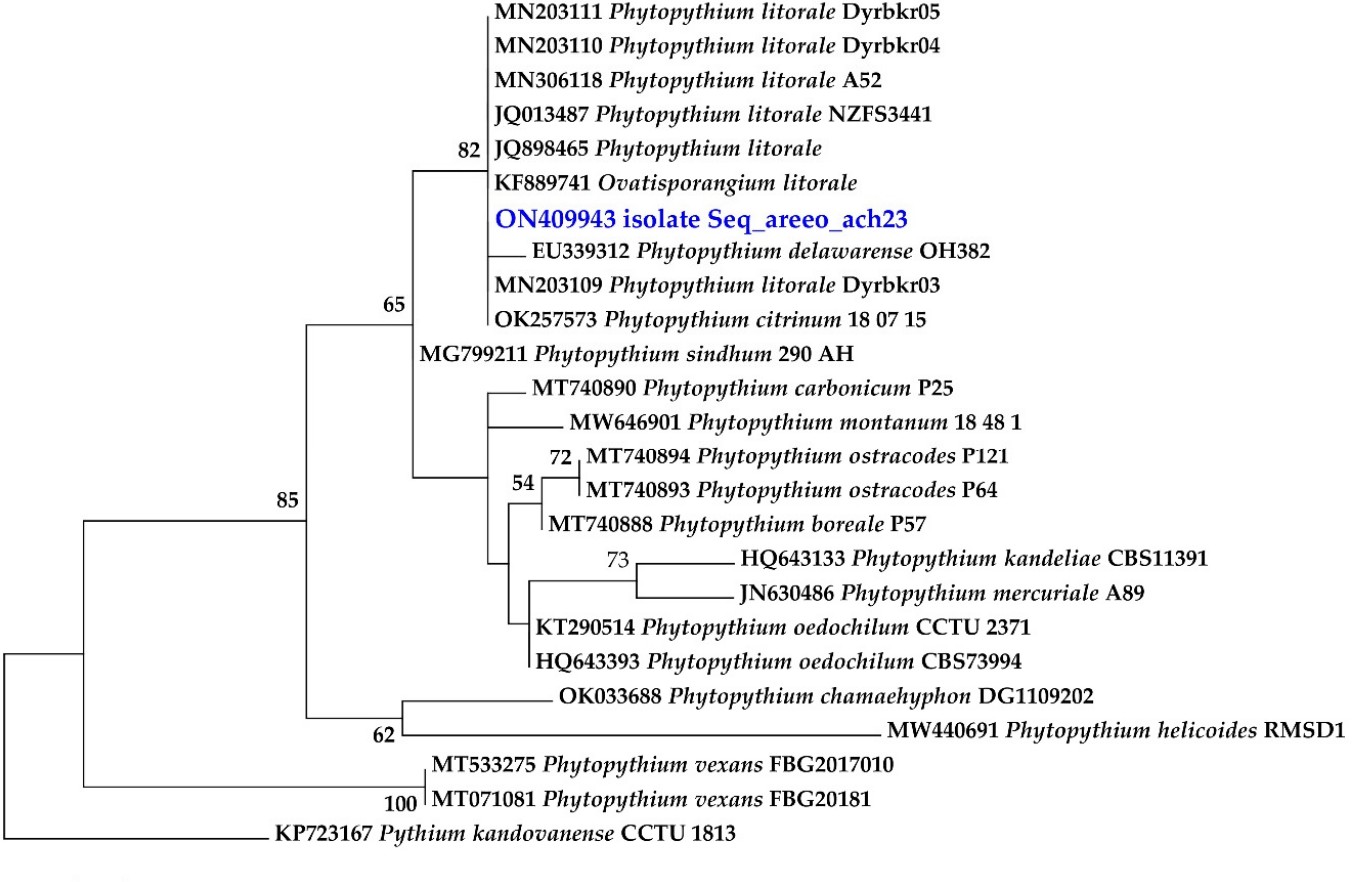

**Figure 2.** Phylogenetic position of Seq_areeo_ach23 isolate (accession number: ON409943) among other *Phytopythium* species based on maximum likelihood analysis of ITS-rDNA sequences. Bootstrap values of more than 50% from 1000 replications are shown on respective branches. *Pythium kandovanense* (CCTU 1813) was used as outgroup.

### 3.2. Efficacy of VOCs Dilutions against Phytopythium Growth

The PA of almond (dilutions: 1, 1/2 and 1/4), cypress (dilution: 1), pine, pomegranate, and walnut (dilution: 1), and pistachio (dilutions: 1 and 1/2) had the best inhibition rates of *Phytopythium* mycelial growth. Pistachio (D1 and D1/2) and almond (D1/2 and D1/4) treatments produced similar results that were significantly different from the control. Almond, cypress, and pistachio dilution 1 completely inhibited the growth of *Phytopythium* (Figures 3 and 4).

Pistachio, almond, and cypress wood vinegars dilution 1 had the best inhibition rates, and were significantly similar to each other and different from other treatments. In the next stages, pomegranate, walnut, and pine had the best inhibitory effects. Significantly, the walnut treatment had a low inhibition rate of the mycelial growth of *Phytopythium* sp. in comparison with other wood vinegars (Figure 5A). In dilution 1/2, the wood vinegars of pistachio, almond, cypress, pomegranate, walnut, and pine had effective inhibitory effects against mycelial growth of *Phytopythium*. Pomegranate and cypress treatments had the same impact (Figure 5A). In dilution 1/4, treatments using the PA of pistachio, almond, pomegranate, walnut, cypress, and pine had the strongest to lowest impact on growth of *Phytopythium* mycelia, respectively (Figure 5A). In dilution 1/8, the treatments of almond, cypress, pistachio and pomegranate, walnut, and pine proved effective against *Phytopythium* sp., (Figure 5A). In dilution 1/10, we noticed a decrease in the inhibition rate

of *Phytopythium* sp. using almond, pistachio, pomegranate, cypress, walnut, and pine wood vinegars (Figure 5A).

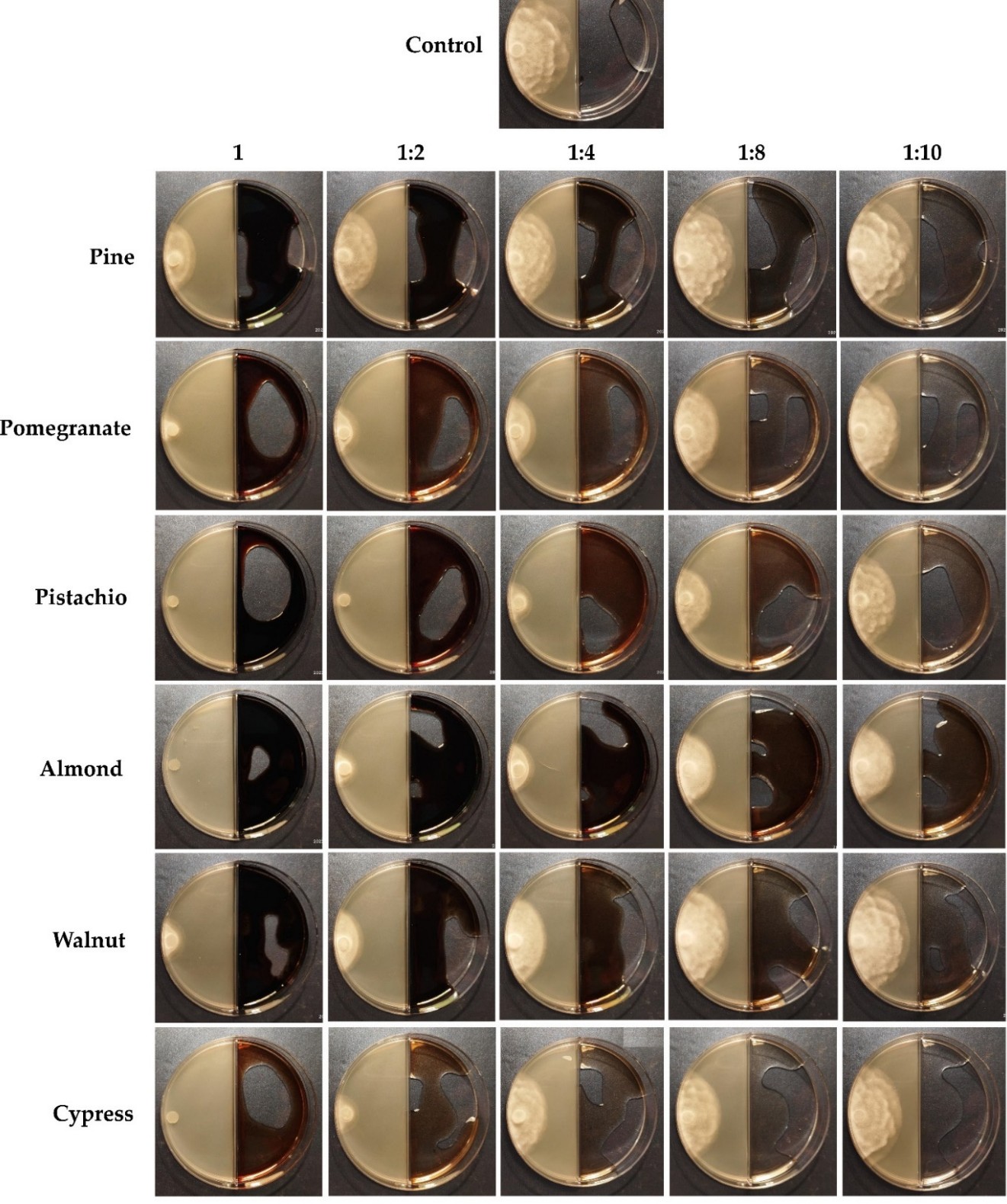

**Figure 3.** In vitro assay of six plant species wood vinegar using 5 dilutions against mycelial growth of *Phytopythium* sp. (ON409943).

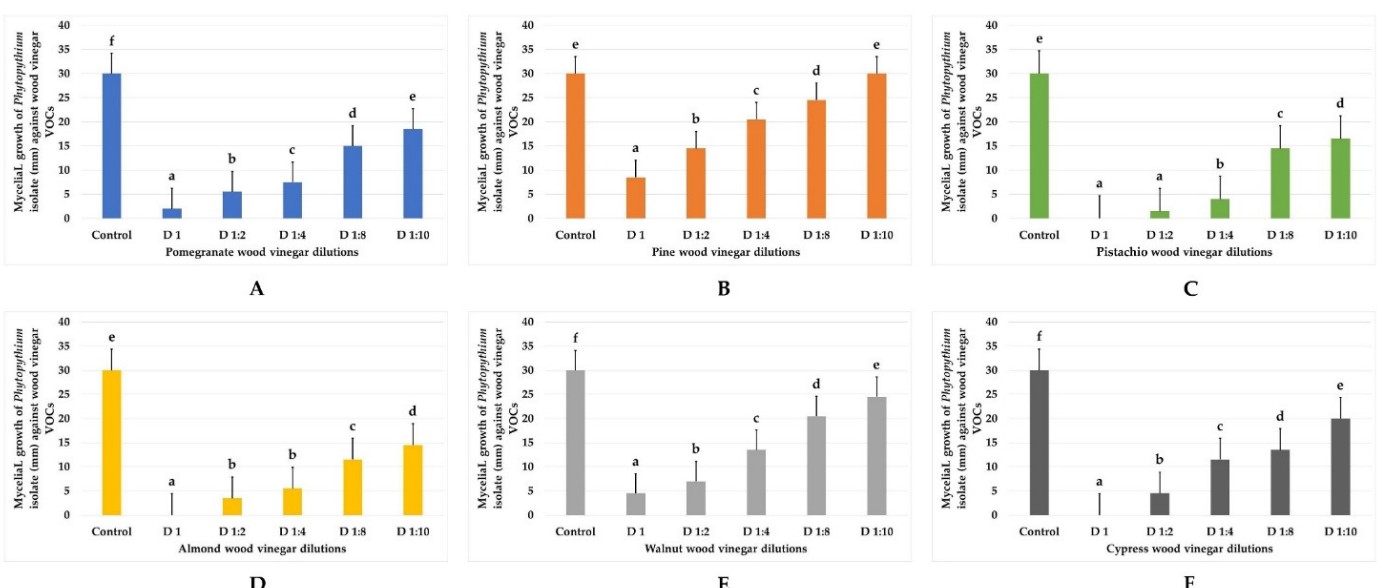

**Figure 4.** Bar charts of effectiveness of different wood vinegars against *Phytopythium* sp. (ON409943). Pomegranate wood vinegar dilutions (**A**); pine wood vinegar dilutions (**B**); pistachio wood vinegar dilutions (**C**); almond wood vinegar dilutions (**D**); walnut wood vinegar dilutions (**E**); cypress wood vinegar dilutions (**F**). Bars labeled with different letters represent significant differences among the treatments at *p* < 0.05 using Tukey's HSD test.

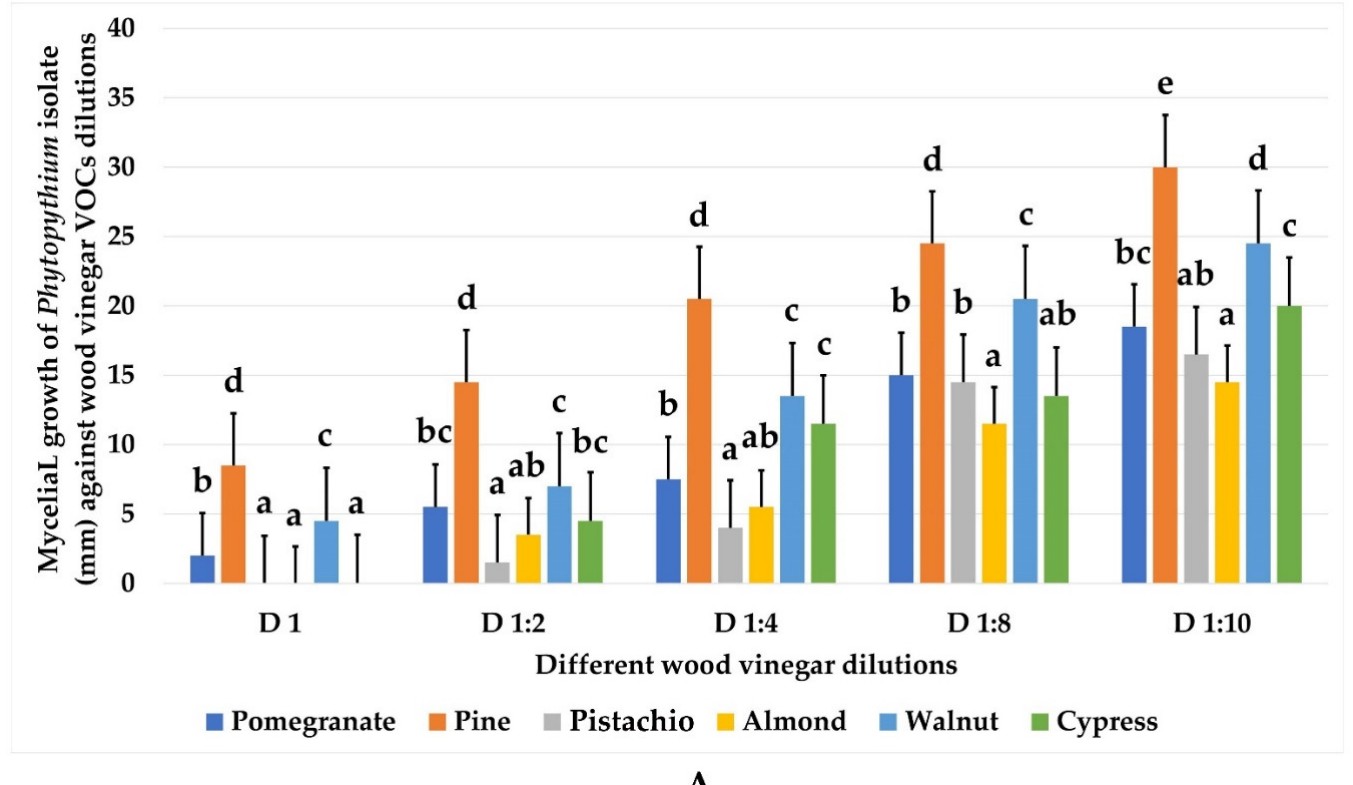

**Figure 5.** *Cont.*

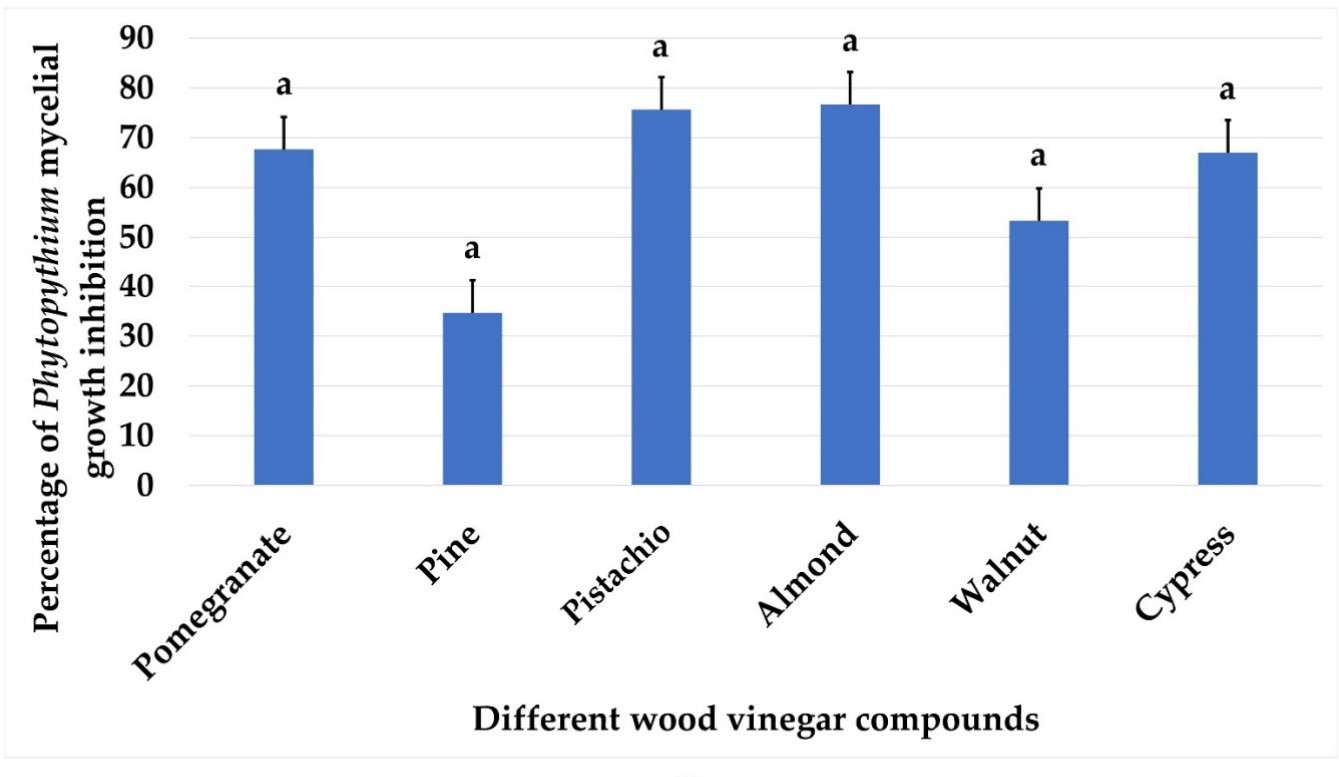

**B**

**Figure 5.** (**A**) Bar chart comparing of different wood vinegar dilutions against mycelial growth of *Phytopythium* sp. (ON409943). (**B**) Percentage of mycelial growth inhibition of *Phytopythium* sp. by different wood vinegars used in this study. Bars labeled with different letters represent significant differences among the treatments at $p < 0.05$ using Tukey's HSD test.

Generally, the comparison of the inhibition rates of wood vinegars toward *Phytopythium* sp. allowed us to conclude that all VOCs significantly decreased the mycelial growth of *Phytopythium* sp. We also concluded that there was no difference in *Phytopythium* sp. percentage of inhibition among them (Figure 5B).

## 4. Discussion

An emerging disease in tomato fields in the province of East-Azerbaijan in Iran was used as the starting point of this research. We were able to isolate the causal agent pathogen and identify its phylogenetic position using a combination of morphological and molecular features. Using the recommendations of Belbahri et al. [18], we avoided the use of environmental and health-damaging synthetic pesticides and managed to find a smart method that combines the use of agricultural biomass waste to treat the disease causal agents giving, therefore, a high added value to these agricultural wastes.

This study showed that the VOCs of wood vinegar recovered from different woody tree materials significantly inhibited the mycelial growth of *Phytopythium* sp. In line with our findings, the antifungal and antioomycete effects of different wood vinegars and their effective usage for management of some plant diseases have been documented [33,49,67,68]. Several scientists unambiguously demonstrated that the antifungal effect of wood vinegar is related to its phenolic compounds [67,69,70]. Guaiacol, 4-ethyl, 2, methoxy phenol, 6-2, dimethoxy phenol, and ethyl acetate are the most important phenolic compounds that have antifungal impacts [71]. Additionally, the existence of phenolic compounds and acetic acid in PA improves its antifungal effect [72,73]. Based on a broad diversity of wood vinegar compounds, it is impossible to consider a single mechanism responsible for its antifungal effects. This finding suggests that several continuous mechanisms are used to provide the observed antimicrobial impact [74]. For instance, in comparison to wood vinegar, a

combination of compost and wood vinegar had the ability to decrease up to 67% of the symptoms of muskmelon root rot caused by *Monosporascus cannonbalus* [74]. Recent reports targeting the antibacterial, antifungal, and antioomycete wood vinegar effects highlighted the huge potential of wood vinegar for controlling plant pests and diseases [75]. This resulted in a significantly reduced disease incidence of many pests in greenhouse and field conditions [76].

Wood vinegar is a low-price material, and Grewal et al. [21] suggested that its price represents only one-third of the cost of synthetic fungicides. It also has the advantage of being a recycled material that can be efficiently used in organic agriculture, in contrast with synthetic pesticides that are prohibited. Moreover, wood vinegar was efficiently used in tomato seed priming for the mitigation of abiotic stresses, and allowed efficient germination and seedling growth [77].

Additionally, wood vinegar was shown to improve the growth, yield, and quality of diverse crops [78], as well as the abiotic stress tolerance of these crops [79]. Wood vinegar has a positive impact on plant growth mainly through the presence of methanol and furfural compounds [80]. Esters compounds can increase chlorophyll and stimulate photosynthesis, and may help sugar and amino acid production, allowing increased resistance of plants to pests and pathogens [55]. Wood vinegar consists of 15 important elements including Na, Al, Mn, K, Ca, Fe, Cd, Cr, Cu, As, P, Pb, Zn, and Mo [36] that have key roles in plant life cycles and that increase photosynthesis. Among them, Fe is one of the most important elements that is a part of all enzymes and the oxidation-reduction reactions necessary for chlorophyll synthesis [81]. The coexistence of acetic acid with cations may cause a complex solution where the ionic bond is replaced with a covalent bond, preventing Fe sedimentation and the water drainage of other elements [81–83]. Therefore, in the next step, the identification of the effective volatile compounds of wood vinegars used in the current study is strongly suggested.

Wood vinegar was also shown to improve soil physico-chemical parameters [82] as well as microbiome content, including plant-growth-promoting rhizobacteria [83]. Actually, its crop growth promoting properties are partly linked to its ability to improve rhizosphere chemical properties and regulate the bacterial community [83].

Future experiments will include the use of wood vinegars in in vivo experiments using a tomato–*Ovatisporangium* sp. patho-system. We also intend to check whether wood vinegar can be applied as a general strategy to cope with oomycete diseases in other patho-systems.

## 5. Conclusions

In this study, we tackled a local problem concerning the emergence of new emerging oomycete disease in tomato fields in the province of East-Azerbaijan in Iran. We identified the causal agent of tomato root rot disease and developed environmentally friendly means of coping with this disease. Our results clearly documented the usage of wood vinegar to manage the disease. Wood vinegar, in addition to being a means of dealing with agricultural biomass waste, allows control of *Ovatisporangium* sp. Our strategy provides, therefore, an environmentally sound and sustainable practice toward the cleaner production of tomatoes. We suggest the use of wood vinegar as a general strategy to cope with oomycete diseases.

**Author Contributions:** Conceptualization, A.C.B., A.N., A.A. and A.T.; methodology, A.C.B., S.N., K.S. and L.B.; software, A.C.B. and A.N.; validation, A.C.B., A.N. and L.B.; formal analysis, A.C.B., A.A., G.E. and L.B.; investigation, A.C.B., G.E., A.T., A.A., K.S. and S.N.; resources, A.C.B., T.O. and L.B.; data curation, A.C.B.; writing—original draft preparation, A.C.B. and L.B.; writing—review and editing, A.C.B., A.N. and L.B.; visualization, A.C.B. and F.N.A.; supervision, A.C.B., K.S., S.N. and L.B.; project administration, A.C.B.; funding acquisition, A.T. and A.C.B. All authors have read and agreed to the published version of the manuscript.

**Funding:** This paper is provided from a part of the project financially supported by Iranian Plant Protection Research Institute (Project No. 2-35-16-004-000029).



**Acknowledgments:** We would like to thank Saeid Ghasemi (University of Zanjan, Iran), Davoud Shirdel (AREEO, Iran), and Motoaki Tojo (Osaka Prefecture University, Japan) for their technical assistance.

**Conflicts of Interest:** The authors declare no conflict of interest.

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
