# Peer review of "In Vitro Evaluation of Wood Vinegar (Pyroligneous Acid) VOCs Inhibitory Effect against a Fungus-like Microorganism Ovatisporangium (Phytopythium) Isolate Recovered from Tomato Fields in Iran"

_agronomy, doi:10.3390/agronomy12071609_

Round 1
Reviewer 1 Report
Needs major improvement in design and presentation, and kindly provide supplementary material for data as well.
Author Response
Reviewer 1
Open Review
(x) I would not like to sign my review report
( ) I would like to sign my review report
English language and style
( ) Extensive editing of English language and style required
(x) Moderate English changes required
( ) English language and style are fine/minor spell check required
( ) I don't feel qualified to judge about the English language and style
|
Yes |
Can be improved |
Must be improved |
Not applicable |
|
|
Does the introduction provide sufficient background and include all relevant references? |
( ) |
( ) |
(x) |
( ) |
|
Are all the cited references relevant to the research? |
( ) |
(x) |
( ) |
( ) |
|
Is the research design appropriate? |
(x) |
( ) |
( ) |
( ) |
|
Are the methods adequately described? |
(x) |
( ) |
( ) |
( ) |
|
Are the results clearly presented? |
( ) |
(x) |
( ) |
( ) |
|
Are the conclusions supported by the results? |
( ) |
( ) |
(x) |
( ) |
Comments and Suggestions for Authors
Corresponding Author: Many thanks for reviewer constructive comments. We tried to address all of them to provide the best quality of the manuscript.
Needs major improvement in design and presentation, and kindly provide supplementary material for data as well.
Corresponding Author: Major changes have been conducted and we trust the manuscript have dramatically improved. We do not have any Supplementary material for data.
Reviewer 2 Report
1. The abstract should be rewritten to summarize the paper. The abstract should briefly present the purpose of the research, the main results and the important coclusions.
2. The introduction should show the uniqueness and advantages, which is the novelty of this paper compared to previous research. I suggest impruving the introduction.
3.The content in many places were only listed, being insufficiently organized and discussed.
4. I do not see the conclusions of the paper!
Author Response
Reviewer 2
Open Review
( ) I would not like to sign my review report
(x) I would like to sign my review report
English language and style
( ) Extensive editing of English language and style required
(x) Moderate English changes required
( ) English language and style are fine/minor spell check required
( ) I don't feel qualified to judge about the English language and style
|
Yes |
Can be improved |
Must be improved |
Not applicable |
|
|
Does the introduction provide sufficient background and include all relevant references? |
( ) |
(x) |
( ) |
( ) |
|
Are all the cited references relevant to the research? |
( ) |
(x) |
( ) |
( ) |
|
Is the research design appropriate? |
( ) |
(x) |
( ) |
( ) |
|
Are the methods adequately described? |
(x) |
( ) |
( ) |
( ) |
|
Are the results clearly presented? |
( ) |
(x) |
( ) |
( ) |
|
Are the conclusions supported by the results? |
( ) |
( ) |
( ) |
(x) |
Comments and Suggestions for Authors
Corresponding Author: Many thanks for reviewer constructive comments. We tried to address all of them to provide the best quality of the manuscript.
- The abstract should be rewritten to summarize the paper. The abstract should briefly present the purpose of the research, the main results and the important coclusions.
Corresponding Author: The manuscript have been rewritten and now briefly present the purpose of the research, the main results and the important conclusions.
- The introduction should show the uniqueness and advantages, which is the novelty of this paper compared to previous research. I suggest impruving the introduction.
Corresponding Author: The introduction section has been dramatically improved. The introduction shows now in its new version the uniqueness and advantages as well as the novelty of this paper compared to previous research.
3.The content in many places were only listed, being insufficiently organized and discussed.
Corresponding Author: The content has now been improved. It is sufficiently organized and discussed.
- I do not see the conclusions of the paper!
Corresponding Author: A conclusion section have been added now to the manuscript.
We trust we performed all changes requested by the reviewers and believe our manuscript is ready for acceptance.
Best regards
Ali Chenari Bouket
Round 2
Reviewer 1 Report
The updated MS is fine, while one thing is still in question about the %Inhibitation data, they are at par to each other. It may happen due to the minimum differences in data and also statistically not sigificant. Is it so??
Author Response
Open Review
( ) I would not like to sign my review report
(x) I would like to sign my review report
English language and style
( ) Extensive editing of English language and style required
( ) Moderate English changes required
(x) English language and style are fine/minor spell check required
( ) I don't feel qualified to judge about the English language and style
|
Yes |
Can be improved |
Must be improved |
Not applicable |
|
|
Does the introduction provide sufficient background and include all relevant references? |
(x) |
( ) |
( ) |
( ) |
|
Are all the cited references relevant to the research? |
(x) |
( ) |
( ) |
( ) |
|
Is the research design appropriate? |
(x) |
( ) |
( ) |
( ) |
|
Are the methods adequately described? |
(x) |
( ) |
( ) |
( ) |
|
Are the results clearly presented? |
( ) |
(x) |
( ) |
( ) |
|
Are the conclusions supported by the results? |
(x) |
( ) |
( ) |
( ) |
Corresponding Author: We appreciate reviewers. We checked English of paper and did some changes, corrections and modifications to improve it.
Comments and Suggestions for Authors
The updated MS is fine, while one thing is still in question about the %Inhibitation data, they are at par to each other. It may happen due to the minimum differences in data and also statistically not sigificant. Is it so??
Corresponding Author: Description of this part is improved now in the paper. Briefly, we mean that all of wood vinegar treatments are significantly effective against Phytopythium sp. but there is not any difference among them (wood vinegar treatments) in percentage of inhibition of oomycete mycelial growth. All of treatments could inhibit the mycelial growth without any significant difference among them.
This manuscript is a resubmission of an earlier submission. The following is a list of the peer review reports and author responses from that submission.